# ADMM-Net for Beamforming Based on Linear Rectification with the Atomic Norm Minimization

**Zhenghui Gong, Xinyu Zhang, Mingjian Ren \*, Xiaolong Su**  **and Zhen Liu** 

College of Electronic Science and Technology, National University of Defense Technology, Changsha 410073, China; gongzhenghui10@nudt.edu.cn (Z.G.); zhangxinyu90111@163.com (X.Z.); zhen_liu@nudt.edu.cn (Z.L.)
\* Correspondence: renmingjian97@163.com

**Abstract:** Target misalignment can cause beam pointing deviations and degradation of sidelobe performance. In order to eliminate the effect of target misalignment, we formulate the jamming subspace recovery problem as a linearly modified atomic norm-based optimization. Then, we develop a deep-unfolding network based on the alternating direction method of multipliers (ADMM), which effectively improves the applicability and efficiency of the algorithm. By using the back-propagation process of deep-unfolding networks, the proposed method could optimize the hyper-parameters in the original atomic norm. This feature enables the adaptive beamformer to adjust its weight according to the observed data. Specifically, the proposed method could determine the optimal hyper-parameters under different interference noise matrix conditions. Simulation results demonstrate that the proposed network could reduce computational cost and achieve near-optimal performance with low complexity.

**Keywords:** beamforming; alternating direction method of multipliers; atomic norm minimization; deep-unfolding

## 1. Introduction

Adaptive beamforming technology is a extensively employed methodology in various domains, including radar, sonar, wireless communication, and medical imaging. This technology leverages the spatial information dimension inherent in sensor arrays to mitigate interference, clutter, and extraneous signals, consequently enhancing the proficiency of target detection and tracking [1]. With the ongoing advancement in the exploration of beamforming under non-ideal conditions, its practical application in engineering is progressively evolving [2–6].

Typically, adaptive beamforming technology is designed based on certain criteria, such as Minimum Variance Distortionless Response (MVDR), Minimum Mean Square Error (MMSE), and Maximum Signal-to-Noise Ratio (SNR). Nevertheless, conventional adaptive beamforming techniques encounter the following challenges:

(1) Target signals and interference signals often occur simultaneously, making it difficult to distinguish. The presence of target signals can induce deviation in the formed beam from its intended direction, diminishing the effectiveness of interference suppression from sidelobes. In severe instances, this phenomenon may lead to the cancellation of target signals.

(2) Target misalignment caused by biases in target prior information or array structural errors can also cause beam pointing deviations, degradation of sidelobe performance, and self-cancellation of target signals [7–9].

To tackle these challenges, researchers have proposed numerous robust adaptive beamforming methods [10]. The MVDR beamformer incorporates diagonal loading of the sample matrix inversion method, which modifies the relative sizes of the eigenvalues of the sample matrix through diagonal loading, thereby enhancing the robustness of the MVDR

beamforming method [11–14]. These methods, along with another class known as worst-case optimization beamformers, share a degree of equivalence. Worst-case optimization beamformers enhance the robustness of the beamformer against the two aforementioned non-ideal factors by setting a protection range around the intended beam direction. The main problem with this approach is that it is difficult to determine the optimal protection range or optimal diagonal loading parameters. Since the method employs a relaxation approach during the solution process, this cannot guarantee its optimal performance [15–18]. Linearly constrained minimum variance (LCMV) beamformers introduce additional linear constraints during the optimization process to augment the robustness of beamforming. When the guiding vector under the linear constraints exhibits a strong correlation with the actual target guiding vector, this class of methods demonstrates robust performance. However, their drawback is the higher sidelobe deformations [19–24]. Another class of robust adaptive beamforming methods is based on subspace techniques. These methods project the specified guiding vector onto the signal and interference subspaces of the sample covariance matrix, which provides good robustness against guiding vector misalignment [25,26]. By analyzing the projection of the target guiding vector, the subspace-based methods can solve the problem of target signal cancellation caused by the contamination of training samples. Nevertheless, these methods necessitate prior knowledge of the target number, a requirement that poses challenges under conditions characterized by a low signal-to-noise ratio and a high-dimensional signal–interference subspace. In addition to the aforementioned traditional methods, recent research has started to use subspace reconstruction methods for adaptive beamforming [27–30]. Sparse recovery methods are employed to recover the interference subspace matrix and calculate adaptive weights based on the reconstructed interference subspace [27], which utilizes the low-rank characteristics of the interference or clutter subspace in the entire sample matrix. However, the above-mentioned methods require the guiding vectors of interference and clutter to fall onto the dictionary matrix of sparse recovery, which cannot handle off-grid situations. Moreover, this class of methods also utilizes a protection range to mitigate the impact of target signals, thereby necessitating prior knowledge of the target protection range. In order to enhance the robustness of adaptive beamforming in the aforementioned two scenarios, a robust adaptive beamforming method in [31] is proposed with the linearly modified atomic norm-based optimization, which utilizes the linearly modified atomic norm-based optimization algorithm to simultaneously estimate the target guiding vector and reconstruct the interference subspace. Herein, the atomic norm-based optimization method does not necessitate prior knowledge of the target and interference guiding vectors. Furthermore, this method separates the interference subspace from the data, which can approach the optimal output signal-to-noise ratio performance [32–34]. However, this method also requires the determination of hyper-parameters during the solution process, and the choice of hyper-parameters has a significant impact on the performance of the method [35–39].

To address this problem, this paper proposes a robust adaptive beamforming method based on deep-unfolding networks, which adopts the form of deep-unfolding networks for the subspace recovery algorithm based on atomic norm optimization, and uses the back-propagation process of deep-unfolding networks to optimize the hyper-parameters in the traditional atomic norm optimization iteration. Furthermore, the proposed method determines the optimal hyper-parameters under varying interference noise matrix conditions.

The remainder of this paper is organized as follows: Section 2 presents the signal model, Section 3 introduces the unfolded algorithm based on linearly corrected atomic norm minimization with ADMM, Section 4 analyzes the experimental results, and Section 5 provides a comprehensive summary of the entire paper.

## 2. Signal Model

Considering a linear array with $M$ omnidirectional antennas, the received signal can be expressed as

$$\mathbf{Y} = \mathbf{S} + \mathbf{J} + \mathbf{N} \tag{1}$$

where $\mathbf{Y} = [y(1), y(2), \ldots, y(T)]$ denotes the received signal with $T$ snapshots, $\mathbf{S}$, $\mathbf{J}$ and $\mathbf{N}$ denote the target signal, interference and Gaussian white noise, respectively. The signals are assumed to be independent of each other at different time periods; the array manifold can be expressed as $\mathbf{a}(\theta) = (1/\sqrt{M})[1, e^{j(2\pi d/\lambda)\sin(\theta)}, \ldots, e^{j(2\pi d/\lambda)(M-1)\sin(\theta)}]^T$, where $d$ denotes element interval, $\lambda$ denotes the wavelength of radar, $(\cdot)^T$ denotes transpose operator.

Herein, the target signal and interference signal can be respectively expressed as

$$\mathbf{S} = c_0 \mathbf{a}(\theta_0) \mathbf{b}_0^T \tag{2}$$

$$\mathbf{J} = \sum_{i=1}^{K} c_i \mathbf{a}(\theta_i) \mathbf{b}_i^T \tag{3}$$

where $\theta_0$ denotes the DOA of target signal, which can be obtained using existing algorithms [40,41]. $\theta_i$ denotes the DOA of the $i$th interference signal, and $\mathbf{b}_0 = [\mathbf{b}_0(1), \mathbf{b}_0(2), \ldots, \mathbf{b}_0(T)]^T$ and $\mathbf{b}_i = [\mathbf{b}_i(1), \mathbf{b}_i(2), \ldots, \mathbf{b}_i(T)]^T$, respectively, denote the normalized complex amplitudes of the target and interference. $c_i$ denotes the real positive value of signal power. Herein, the interference signal power is much greater than the target signal power. Moreover, the mean and variance of target signal are zero and $\sigma^2$, which is independent of the inference signal and noise.

Adaptive digital beamforming (ADBF) is designed to eliminate the interference signal by applying adaptively calculated weights for the received signal; the output can be expressed as

$$\overline{\mathbf{Y}}^T = \mathbf{w}^H(\mathbf{S} + \mathbf{J} + \mathbf{N}) \tag{4}$$

where $(\cdot)^H$ denotes the conjugate transpose operator.

In order to suppress interference, $\mathbf{w}$ should be orthogonal to the interference subspace while keeping the main lobe unchanged along the target direction. The classical Wiener filter determines the weights by solving the following linear-constrained quadratic optimization, which can be expressed as

$$\min_{\mathbf{w}} \mathbf{w}^H \mathbf{R} \mathbf{w}, \ \text{s.t.} \mathbf{w}^H \mathbf{a}(\theta) = 1 \tag{5}$$

where $\theta$ represents the expected pointing direction of the adaptive beam, $\mathbf{R} = \mathbb{E}(\mathbf{Y}\mathbf{Y}^H)$ represents the data covariance matrix, $\mathbb{E}(\cdot)$ represents the calculated statistical expectation. Since the estimated covariance $\hat{\mathbf{R}}$ also includes the target signal, the adaptive beamformer can also offset the target signal.

As for the data of the target contamination, $\mathbf{w}$ should be calculated as

$$\min_{\mathbf{w}} \mathbf{w}^H \mathbf{R}_J \mathbf{w}, \ \text{s.t.} \mathbf{w}^H \mathbf{a}(\theta) = 1 \tag{6}$$

where $\mathbf{R}_J$ is a Hermitian Toeplitz matrix and can be considered as the interference signal subspace, i.e., $\mathbf{R}_J = \sum_{i=1}^{K} c_i^2 \mathbf{a}(\theta_i) \mathbf{a}(\theta_i)^H$. However, $\mathbf{R}_J$ is very difficult to obtain.

## 3. Proposed Algorithm

### 3.1. ADMM Model Based on the Linear Correction Atomic Norm

Traditional methods recover the interference signal subspace by decomposition of the eigenvalues of $\mathbf{R}$. However, this method is only effective when the number of interference sources is known and the interference-to-noise ratio (INR) is high. Moreover, the performance of the algorithm relies heavily on the estimation of the matrix, which indicated that a large number of snapshots are required for an accurate recovery of the matrix. As well, the conventional methods still need to assume that the target signal subspace is approximately orthogonal to the interference signal subspace. Therefore, we need new methods to estimate $\mathbf{R}_J$.

In this paper, by combining our approach with the start-of-art methods, we build a model based on linear rectification of atomic norms for the beamforming problem in the case of target corrupted training data and use the ADMM algorithm to solve it effectively.

The core of the problem is to find a solution with the least number of atoms to describe **J**, while **S** + **J** is bounded within the Frobenius norm sphere around **Y**. Therefore, the following problem models can be expressed as

$$\min_{\mathbf{X},\theta_0,\mathbf{s}} \| \mathbf{X} - \mathbf{a}(\theta_0)\mathbf{s}^T \|_{\mathcal{A},0} \ \ \text{s.t.} \frac{1}{2} \| \mathbf{Y} - \mathbf{X} \|_{\mathcal{F}}^2 \leq \eta \tag{7}$$

where **X** is the estimate of **S** + **J**, **s** is the estimate of the target signal, $\| \cdot \|_{\mathcal{F}}$ denotes the $F$ norm, $\eta$ denotes the artificial parameters related to the noise power, $\| \cdot \|_{\mathcal{A},0}$ denotes the non-convex norm, which can be defined as

$$\| \mathbf{X} \|_{\mathcal{A},0} = \inf_r \left\{ \mathbf{X} = \sum_i^r c_i \mathbf{A}(\theta_i, \mathbf{b}_i), c_i \geq 0 \right\} \tag{8}$$

where $r$ denotes the number of atoms by forming the interference signal.

Minimizing directly (8) is, however, proven to be NP-hard. Thus, we opt to utilize the atomic norm as a convex relaxation of the atomic $\ell 0$ norm.

$$\min_{\mathbf{X},\theta_0,\mathbf{s}} \| \mathbf{X} - \mathbf{a}(\theta_0)\mathbf{s}^T \|_{\mathcal{A}} \ \ \text{s.t.} \frac{1}{2} \| \mathbf{Y} - \mathbf{X} \|_{\mathcal{F}}^2 \leq \eta \tag{9}$$

where $\| \cdot \|_{\mathcal{A}}$ denotes the atomic norm, which is defined as

$$\| \mathbf{X} \|_{\mathcal{A}} = \inf \left\{ \sum_i c_i \ | \ \mathbf{X} = \sum_i c_i \mathbf{A}(\theta_i, \mathbf{b}_i), c_i \geq 0 \right\} \tag{10}$$

By using the Schur complementary lemma and definitions of atomic norm, the above optimization allows the following equivalent SDP features

$$\min_{\mathbf{X},\mathbf{s}} \inf_{\mathbf{u}\in\mathcal{C}^M,\mathbf{\Omega}\in\mathcal{C}^{T\times T}} \left( \frac{1}{2}\text{Tr}(\tau(\mathbf{u})) + \frac{1}{2}\text{Tr}(\mathbf{\Omega}) \right)$$
$$\text{s.t.} \begin{bmatrix} \tau(\mathbf{u}) & \mathbf{X} - \mathbf{a}(\hat{\theta}_0)\mathbf{s}^T \\ (\mathbf{X} - \mathbf{a}(\hat{\theta}_0)\mathbf{s}^T)^H & \mathbf{\Omega} \end{bmatrix} \succeq 0, \frac{1}{2} \| \mathbf{Y} - \mathbf{X} \|_{\mathcal{F}}^2 \leq \eta \tag{11}$$

where $\tau(\mathbf{u})$ is the Hermitian Toeplitz matrix formed by the vector as the first column, $\mathbf{\Omega}$ is a variable matrix, $\text{Tr}(\cdot)$ denotes the trace of the matrix, which can be utilized to achieve the optimization based on the atomic norm. Note that the semidefinite constraint in (11) means $\mathbf{X} - \mathbf{a}(\hat{\theta}_0)\mathbf{s}^T$ having the same column space as $\tau(\mathbf{u})$. More specifically, $\tau(\mathbf{u})$ is the estimate of the interference subspace $\sum_{i=1}^{K} c_i^2 \mathbf{a}(\theta_i)\mathbf{a}(\theta_i)^H$.

In order to apply the ADMM method, (11) is repeated as

$$\min_{\mathbf{X},\mathbf{s},\mathbf{u},\mathbf{\Omega}} \frac{\varepsilon}{2}(\text{Tr}(\tau(\mathbf{u})) + \text{Tr}(\mathbf{\Omega})) + \frac{1}{2} \| \mathbf{Y} - \mathbf{X} \|_{\mathcal{F}}^2$$
$$\text{s.t.} \mathbf{Z} = \begin{bmatrix} \tau(\mathbf{u}) & \mathbf{X} - \mathbf{a}(\hat{\theta}_0)\mathbf{s}^T \\ (\mathbf{X} - \mathbf{a}(\hat{\theta}_0)\mathbf{s}^T)^H & \mathbf{\Omega} \end{bmatrix}, \mathbf{Z} \succeq 0 \tag{12}$$

where $\varepsilon$ is the regularization parameter related to $\eta$. Firstly, the augmented Lagrangian function of (12) is expressed as

$$\begin{aligned}
\varphi(\mathbf{X}, \mathbf{s}, \mathbf{\Omega}, \mathbf{\Lambda}, \mathbf{Z}) = & \tfrac{1}{2} \parallel \mathbf{X} - \mathbf{Y} \parallel_{\mathcal{F}}^2 + \tfrac{\varepsilon}{2}(\mathrm{Tr}(\varepsilon(\mathbf{u})) + \mathrm{Tr}(\mathbf{\Omega})) \\
& + \left\langle \mathbf{\Lambda}, \mathbf{Z} - \begin{bmatrix} \tau(\mathbf{u}) & \mathbf{X} - \mathbf{a}(\theta_0)\mathbf{s}^T \\ (\mathbf{X} - \mathbf{a}(\theta_0)\mathbf{s}^T)^H & \mathbf{\Omega} \end{bmatrix} \right\rangle \\
& + \tfrac{\rho}{2}\left\Vert \mathbf{Z} - \begin{bmatrix} \tau(\mathbf{u}) & \mathbf{X} - \mathbf{a}(\theta_0)\mathbf{s}^T \\ (\mathbf{X} - \mathbf{a}(\theta_0)\mathbf{s}^T)^H & \mathbf{\Omega} \end{bmatrix} \right\Vert_{\mathcal{F}}^2
\end{aligned} \tag{13}$$

where $\mathbf{\Lambda}$ and $\rho$ are Lagrange multipliers, $\mathbf{Z}$, $\mathbf{\Omega}$ and $\mathbf{\Lambda}$ are Hermite matrices. The update steps for ADMM are as follows

$$\begin{aligned}
(\mathbf{X}^{t+1}, \mathbf{s}^{t+1}, \mathbf{u}^{t+1}, \mathbf{\Omega}^{t+1}) &= \underset{\mathbf{X}, \mathbf{s}, \mathbf{u}, \mathbf{\Omega}}{\arg\min} \varphi(\mathbf{X}, \mathbf{s}, \mathbf{u}, \mathbf{\Omega}, \mathbf{\Lambda}^t, \mathbf{Z}^t) \\
\mathbf{Z}^{t+1} &= \underset{\mathbf{Z} \succeq \mathbf{0}}{\arg\min} \varphi(\mathbf{X}^{t+1}, \mathbf{s}^{t+1}, \mathbf{u}^{t+1}, \mathbf{\Omega}^{t+1}, \mathbf{\Lambda}^t, \mathbf{Z}) \\
\mathbf{\Lambda}^{t+1} &= \mathbf{\Lambda}^t + \rho\left(\mathbf{Z}^{t+1} - \begin{bmatrix} \tau(\mathbf{u}^{t+1}) & \mathbf{X}^{t+1} - \mathbf{a}(\theta_0)(\mathbf{s}^{t+1}) \\ (\mathbf{X}^{t+1} - \mathbf{a}(\theta_0)(\mathbf{s}^{t+1})^H) & \mathbf{\Omega}^{t+1} \end{bmatrix}\right)
\end{aligned} \tag{14}$$

where the superscript $t$ denotes the $t$th iteration.

Moreover, the matrix decomposition in (15) can be expressed as

$$\mathbf{\Lambda} = \begin{bmatrix} \mathbf{\Lambda}_{M \times M} & \mathbf{\Lambda}_{M \times T} \\ \mathbf{\Lambda}_{T \times M} & \lambda_{T \times T} \end{bmatrix}, \mathbf{Z} = \begin{bmatrix} \mathbf{Z}_{M \times M} & \mathbf{Z}_{M \times T} \\ \mathbf{Z}_{T \times M} & \mathbf{Z}_{T \times T} \end{bmatrix} \tag{15}$$

Then, the closed-form update rules can be written as follows:

$$\begin{aligned}
\mathbf{\Omega}^{t+1} &= \mathbf{Z}_{T \times T}^t + \tfrac{1}{\rho}\left(\mathbf{\Lambda}_{T \times T}^t - \tfrac{\varepsilon}{2}\mathbf{I}\right) \\
\mathbf{u}^{t+1} &= \tfrac{1}{\rho}\mathbf{Y}\left(g(\mathbf{\Lambda}_{M \times M}^t) + \rho g(\mathbf{Z}_{M \times M}^t) - \tfrac{\varepsilon}{2}\mathbf{e}_1\right) \\
\overline{\mathbf{X}}^{t+1} &= \left(2\rho\mathbf{F}_1^H\mathbf{F}_1 + \mathbf{F}_2^H\mathbf{F}_2\right)^{-1}\left(\mathbf{F}_2^H\mathbf{Y} + 2\mathbf{F}_1^H\mathbf{\Lambda}_{M \times T}^t + 2\rho\mathbf{F}_1^H\mathbf{Z}_{M \times T}^t\right)
\end{aligned} \tag{16}$$

where $\mathbf{Y}$ is the diagonal matrix with the diagonal element $\mathbf{Y}_{i,i} = \frac{1}{M-i+1}, i = 1, 2, \cdots, M$. $g(\cdot)$ denotes the linear mapping from the matrix to the vector, where the value of the $i$th element corresponds to the sum of the matrix element values satisfied by the number of rows $p$ and columns $q$. $\overline{\mathbf{X}} = [\mathbf{X}^T \mid \mathbf{s}]^T$ denotes that $\mathbf{s}^T$ replaces the $(M + 1)$-th row of $\mathbf{X}$.

By applying eigendecomposition of $\begin{bmatrix} \tau(\mathbf{u}^{t+1}) & \mathbf{X}^{t+1} - \mathbf{a}(\theta_0)(\mathbf{s}^{t+1})^T \\ (\mathbf{X}^{t+1} - \mathbf{a}(\theta_0)(\mathbf{s}^{t+1})^T)^H & \mathbf{\Omega}^{t+1} \end{bmatrix}$ $- \frac{1}{\rho}\mathbf{\Lambda}^t = \sum \sigma_i^t \mathbf{U}_i^t(\mathbf{U}_i^t)^H$, $\mathbf{Z}$ can be rewritten as:

$$\mathbf{Z}^{t+1} = \sum_{i \in \mathcal{D}} \sigma_i^t \mathbf{U}_i^t(\mathbf{U}_i^t)^H \tag{17}$$

where $\mathcal{D} = \{i \mid \sigma_i^t \geq 0\}$.

Based on the closed-form update rules listed in (14), (16), and (17), the solution of the problem is obtained by running the above iterations until a predetermined error tolerance or upper iteration limit is reached.

Since the traditional ADMM algorithm requires hundreds of iterations to obtain the ideal results, the operation efficiency is low. In addition, the parameters $\varepsilon$ and $\rho$ in the algorithm need to be set manually, which leads to a significant impact on the final results. In view of the above problems, this paper combines the ADMM algorithm with the deep-unfolding network to obtain the ADMM network, which effectively improves the applicability and efficiency of the algorithm.

### 3.2. Design of the C-ADMM-Net

The parameters $\mathbf{\Omega}$, $\mathbf{u}$, $\overline{\mathbf{X}}$, $\mathbf{Z}$ and $\mathbf{\Lambda}$ that are involved in the ADMM algorithm require iteration to ensure the correct update of parameters. The initial data can be randomly generated under the Hermitian matrix. Moreover, the guidance vector representing the direction of the target can usually be accurately acquired in advance.

It can be noticed that the final output of the key factors is affected by artificial parameters $\varepsilon$ and $\rho$, which will directly determine the performance and operational complexity of the ADMM algorithm, but with great uncertainty. In total, this paper optimizes the parameter setting for the deep-unfolding network to improve the performance of the algorithm.

#### 3.2.1. The Update Layer of Data

The updated structure of the parameter $\mathbf{\Omega}$ is shown in Figure 1. $\mathbf{\Lambda}$ and $\mathbf{Z}$ are both Hermitian matrices. $\mathbf{\Omega}$ is updated according to (16) with parameters $\varepsilon$, $\rho$ and $Z$.

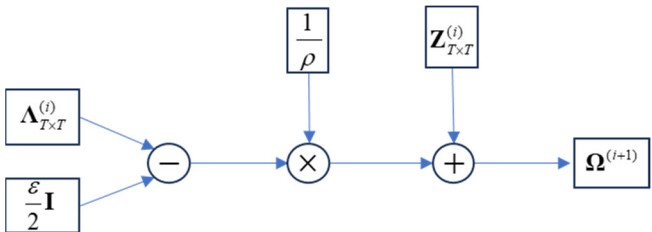

**Figure 1.** The update layer of parameter $\mathbf{\Omega}$.

In the ADMM algorithm, the $i$th iteration needs to artificially set the parameters to $\varepsilon$ and $\rho$. The parameters to be learned in the network are defined as $\varepsilon^{(i)}$ and $\rho^{(i)}$, and the corresponding forward propagation expression can be expressed as

$$\mathbf{\Omega}^{(i+1)} = \mathbf{Z}_{T \times T}^{(i)} + \frac{1}{\rho^{(i)}}\left(\mathbf{\Lambda}_{T \times T}^{(i)} - \frac{\varepsilon^{(i)}}{2}\mathbf{I}\right) \tag{18}$$

Similarly to the update layer of parameter $\mathbf{\Omega}$, the update layer of parameter $\mathbf{u}$ in the network is determined by $\varepsilon^{(i)}$ and $\rho^{(i)}$, a more complex structure. According to the update rule in (16), we can design the structure of the update layer as shown in Figure 2.

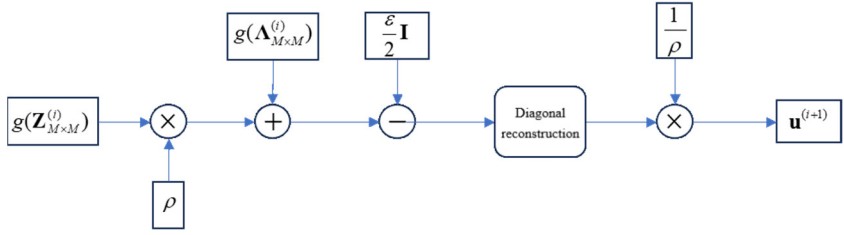

**Figure 2.** The update layer of parameter $\mathbf{u}$.

Accordingly, the corresponding forward propagation is given by

$$\mathbf{u}^{(i+1)} = \frac{1}{\rho}\mathbf{Y}\left(g(\mathbf{\Lambda}_{M \times M}^{(i)}) + \rho g(\mathbf{Z}_{M \times M}^{(i)}) - \frac{\varepsilon}{2}\mathbf{e}_1\right) \tag{19}$$

It can be seen that the parameters $\mathbf{\Omega}$ and $\mathbf{u}$ are only determined by the parameters $\mathbf{Z}$ and $\mathbf{\Lambda}$; $\mathbf{Z}$ and $\mathbf{\Lambda}$ will directly affect the final calculation result.

In addition to the influence of $\mathbf{Z}$ and $\mathbf{\Lambda}$, the update of parameter $\overline{\mathbf{X}}$ also needs to use the received data of the radar. At this time, the only parameter that must be optimized is $\rho^{(i)}$, and the update layer structure is shown in Figure 3.

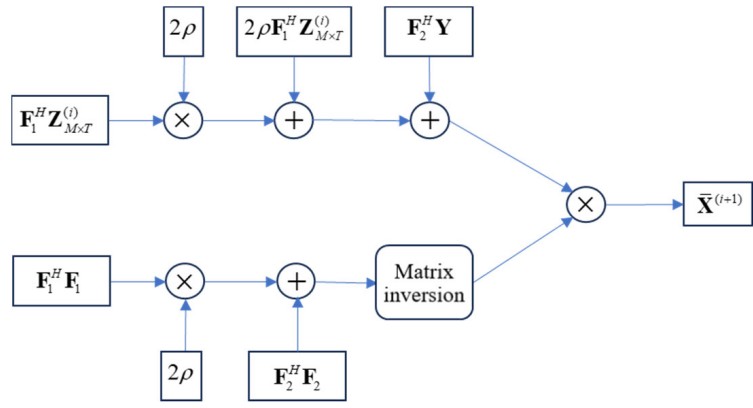

**Figure 3.** The update layer of parameter $\overline{\mathbf{X}}$.

Accordingly, the forward propagation can be expressed as

$$\overline{\mathbf{X}}^{(i)} = \left(2\rho^{(i)}\mathbf{F}_1^H\mathbf{F}_1 + \mathbf{F}_2^H\mathbf{F}_2\right)^{-1}\left(\mathbf{F}_2^H\mathbf{Y} + 2\mathbf{F}_1^H\boldsymbol{\Lambda}_{M\times T}^{(i)} + 2\rho^{(i)}\mathbf{F}_1^H\mathbf{Z}_{M\times T}^{(i)}\right) \tag{20}$$

### 3.2.2. The Update Layer of Matrix Reconstruction

The update of the parameter $\mathbf{Z}$ requires exploiting the results of the output for update layer and parameter $\boldsymbol{\Lambda}$. The calculation process involves matrix reconstruction and data screening. The forward propagation expression can be divided into

$$\begin{bmatrix} \tau(\mathbf{u}^{(i+1)}) & \mathbf{X}^{(i+1)} - \mathbf{a}(\theta_0)(\mathbf{s}^{(i+1)})^T \\ (\mathbf{X}^{(i+1)} - \mathbf{a}(\theta_0)(\mathbf{s}^{(i+1)})^T)^H & \boldsymbol{\Omega}^{(i+1)} \end{bmatrix} - \frac{1}{\rho}\boldsymbol{\Lambda}^{(i)} = \sum \sigma_k^{(i)}\mathbf{U}_k^{(i)}(\mathbf{U}_k^{(i)})^H \tag{21}$$

$$\mathbf{Z}^{(i+1)} = \sum_{k\in\mathcal{D}} \sigma_k^{(i)}\mathbf{U}_k^{(i)}(\mathbf{U}_k^{(i)})^H, \mathcal{D} = \left\{k\left|\sigma_k^{(i)} \geq 0\right.\right\} \tag{22}$$

where $\mathbf{s}^{(i+1)}$ can be obtained from $\overline{\mathbf{X}}^{(i+1)}$. The essence of (22) is to filter the feature data whose feature is greater than or equal to 0. The specific updated structure is shown in Figure 4.

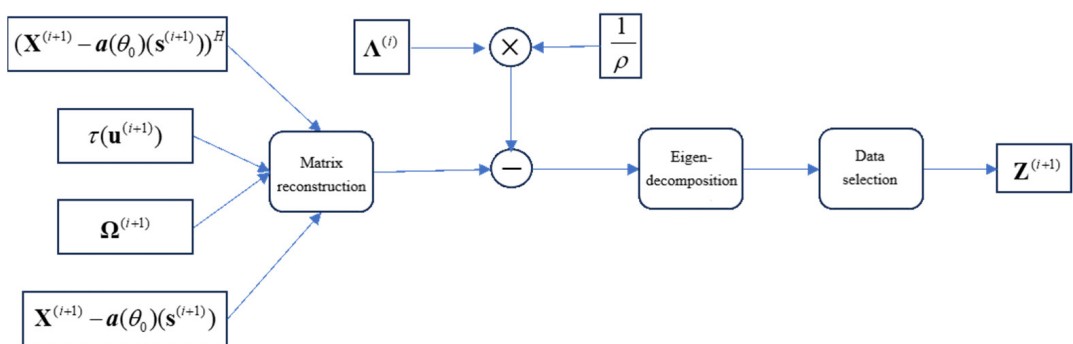

**Figure 4.** The update layer of parameter $\mathbf{Z}$.

The last parameter $\boldsymbol{\Lambda}$ that needs to be updated also needs to be utilized by the matrix reconstruction, which will use all the four parameters that appear in the above update process. At this time, the parameter to be optimized is $\rho^{(i)}$, and the update layer of parameter $\boldsymbol{\Lambda}$ is shown in Figure 5.

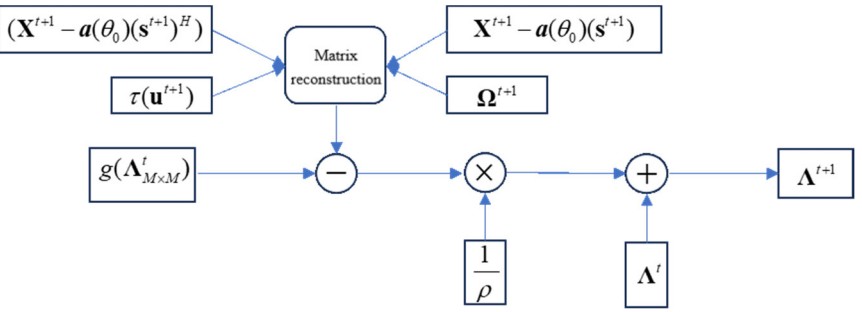

**Figure 5.** The update layer of parameter $\mathbf{\Lambda}$.

Accordingly, the corresponding forward propagation can be expressed as

$$\mathbf{\Lambda}^{(i+1)} = \mathbf{\Lambda}^{(i)} + \rho \left( \mathbf{Z}^{(i+1)} - \begin{bmatrix} \tau(\mathbf{u}^{(i+1)}) & \mathbf{X}^{(i+1)} - \mathbf{a}(\theta_0)(\mathbf{s}^{(i+1)}) \\ (\mathbf{X}^{(i+1)} - \mathbf{a}(\theta_0)(\mathbf{s}^{(i+1)})^H) & \mathbf{\Omega}^{(i+1)} \end{bmatrix} \right) \quad (23)$$

### 3.2.3. Analysis of C-ADMM-Net Structure

The proposed C-ADMM-net consists of a cascade network of an input layer, an output layer, and a P-level substructure, where the output layer is composed of a single update layer of data.

The C-ADMMN network structure is shown in Figure 6. The input layer includes the initial random Hermitian matrices $\mathbf{\Lambda}^0$ and $\mathbf{Z}^0$, as well as the received signal Y. The output layer only needs to calculate the parameter $\mathbf{u}^{\text{out}}$. The above C-ADMM-net contains P update layers of data and P update layers of matrix reconstruction.

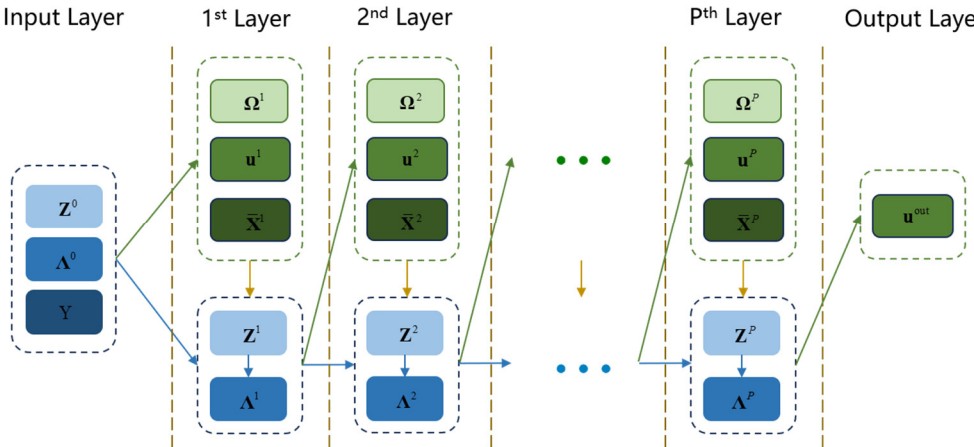

**Figure 6.** The C-ADMM-net Structure.

For this network structure, the parameters to be learned can be expressed as the following sets:

$$\alpha = \left\{ \rho^{(i)}, \varepsilon^{(i)} \middle| i = 0, 1, 2, \dots, P-1 \right\} \quad (24)$$

It can be noticed that two parameters need to be learned for the *P*-level C-ADMM-net. In traditional algorithms, the parameters can be only manually adjusted for the output. However, by using the C-ADMM-net for application, it is necessary to train a more theoretical optimal value and array signal, and then process the received array signal, which can effectively reduce the operational complexity and improve the adaptability.

### 3.2.4. Back Propagation Algorithm in Complex Number Domain

The input of the C-ADMM-net is the array signal and the output is adaptive beamforming weights. The training set of the network can be expressed as $\Theta = \left\{ Y_i, S_i^{label} \right\}_{i=1}^{N_c}$,

where $Y_i$ denotes the $i$th group of array signals, $S_i^{label}$ denotes the corresponding label of SINR, $N_c$ denotes the total number of datasets. In this paper, the SINR corresponding to the theoretical optimal weight is considered as the label, and the loss function is defined as

$$Loss = \frac{1}{N_c} \sum_{i=1}^{i=N_c} \left\| S_i - S_i^{label} \right\|_2 \tag{25}$$

where $\|\cdot\|_2$ denotes the second norm, $S_i$ denotes the SINR obtained from training. The loss function can intuitively reflect the difference between the trained network and the ideal case. By applying (25) and the complex domain BP algorithm, the derivative of $f(\mathbf{O})$ about $\mathbf{O}$ for any complex domain matrix $\mathbf{O}$ and real function $f(\mathbf{O})$ can be calculated as

$$\mathrm{Grad} f(\mathbf{O}) = 2\frac{\mathrm{d}f(\mathbf{O})}{\mathrm{d}\mathbf{O}^*} = \frac{\partial f(\mathbf{O})}{\partial \mathrm{Re}\{\mathbf{O}\}} + j\frac{\partial f(\mathbf{O})}{\partial \mathrm{Im}\{\mathbf{O}\}} \tag{26}$$

where $\mathrm{Re}\{\mathbf{O}\}$ and $\mathrm{Im}\{\mathbf{O}\}$ respectively denote the real and imaginary parts of the matrix. The scalar form of the calculated chain rule of the complex number domain gradient can be further obtained as

$$\frac{\partial f(\eta)}{\partial \eta} = \left\langle \frac{\partial f}{\partial \mathrm{Re}\{\mathbf{O}\}}, \frac{\partial \mathrm{Re}(\mathbf{O})}{\partial \eta} \right\rangle + \left\langle \frac{\partial f}{\partial \mathrm{Im}\{\mathbf{O}\}}, \frac{\partial \mathrm{Im}(\mathbf{O})}{\partial \eta} \right\rangle \tag{27}$$

where $\eta$ denotes the real number scalar, $f(\eta)$ denotes the real-valued function of $\eta$.

By applying the chain rule shown in (27) to the ADMM-Net, the loss function can calculate the gradient of any parameter in the parameter set. After obtaining the gradient, the training can be updated by using the gradient descent.

## 4. Computer Simulation Experiments

### 4.1. Introduction of the Dataset

The experimental section primarily employs various simulation data to verify the algorithm. In the simulation data, the radar is assumed to be a uniform array of 10 units at half wavelength. There are two strong disturbances, incident into the radar from two different directions far away from the main lobe, with the main lobe pointing in the desired direction.

The simulation dataset has a total of 600 sets of array signal data, and the corresponding label optimal SINR, where 300 groups were randomly taken as training samples, leaving 300 groups as testing data. Each set of data contains 10 array echo data; each array echo information group is a $10 \times T$ matrix, where $T$ denotes the number of snapshots, and the number of layers of the expanded network is set to 30 layers.

The specific training parameters are set as follows: the root mean square (RMS) error function is chosen as the loss function, and the utilized optimizer is Adam with typical parameter values of betas = (0.9, 0.999) and eps = $10^{-8}$. Additionally, the learning rate is configured as 0.04. Due to the limited parameters to be learned in the network, the small dataset adopted in the experiment can effectively learn the parameters without producing an overfitting phenomenon.

### 4.2. Experimental Results and Analysis

#### 4.2.1. Contrast of Beamforming Optimization

In scenario 1, the target is located at $\theta_0 = -15°$, two strong disturbances are located $\theta_1 = 10.5°$ and $\theta_2 = 30°$, and the number of snapshots for the DBF collection is 20. The SNR is 0 dB, and the interference-to-noise ratio is 20 dB. The results of the beamforming direction diagram are shown in Figure 7.

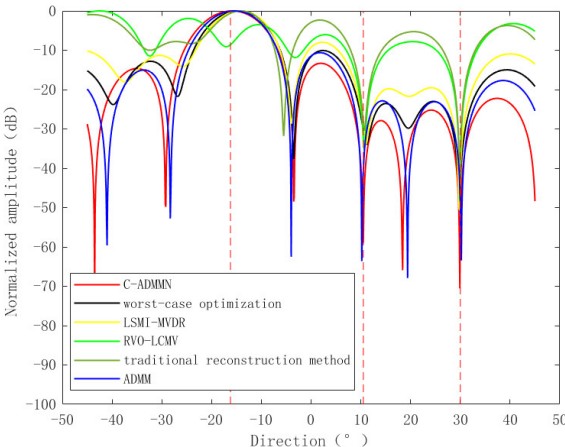

**Figure 7.** The beam patterns of the different algorithms in Scenario 1.

The simulation results indicate that all the algorithms successfully suppress interference; however, the strong RVO-LCMV algorithm exhibits a significant offset in the main lobe, and the traditional reconstruction method algorithm demonstrates notably high side lobes. The ADMM algorithm and the proposed C-ADMMN algorithm exhibit the ability to preserve favorable main and side lobe characteristics while effectively suppressing interference. In comparison, the C-ADMMN algorithm, being closer to the interference direction, demonstrates superior performance.

In Scenario 2, consider cases where the SNR is already high enough for detection. As the target signal becomes stronger, the probability of target self-elimination becomes higher. When the SNR is set to 10 dB, with all other parameters held constant as in scenario 1, the results of the beamforming direction diagram are illustrated in Figure 8.

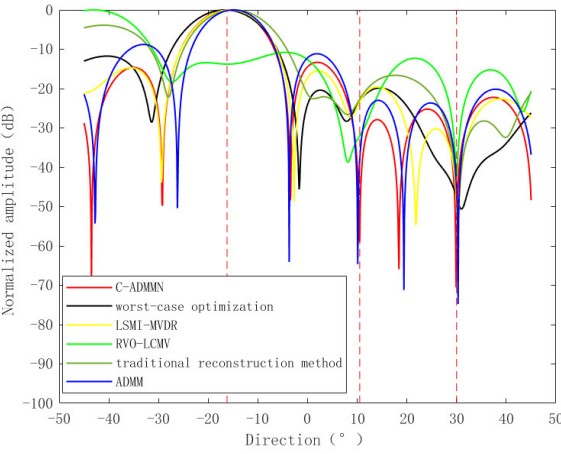

**Figure 8.** Beam patterns of different algorithms in scenario 2.

It can be seen that in this case, all methods avoid target self-elimination. The main value of RVO-LCMV showed a significant deviation, and the side lobe was also higher. Only the ADMM algorithm and the proposed C-ADMMN algorithm can produce a depression at $10.5°$, when the zero-trap direction of the C-ADMMN algorithm is still closer to the interference direction. Subsequently, the investigation delves into scenarios involving variations in target and interference.

In scenario 3, the target direction is changed to $\theta_0 = 10°$, the interference directions are changed to $\theta_1 = 35°, \theta_2 = -16.2°$. The beam direction diagram of the different algorithms is shown in Figure 9. In this case, the proposed algorithm C-ADMMN still has a lower side lobe and a deeper zero point than the traditional algorithm.

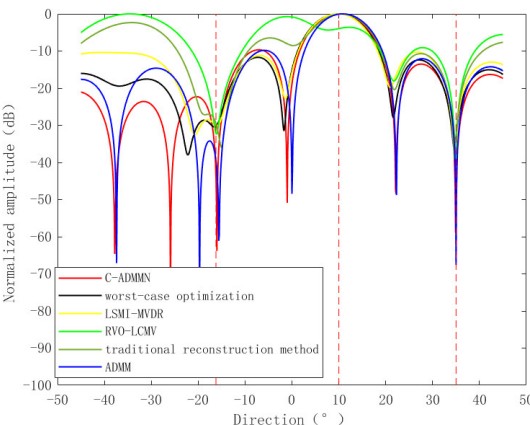

**Figure 9.** Beam patterns of different algorithms in scenario 3.

The test loss function of the three algorithm changes with the number of training times is shown in Figure 10. From the initial training rounds, it is evident that the algorithm's performance with ADMM is suboptimal due to the small number of network layers in conjunction with manually set parameters. However, as the training progresses, the parameters are effectively adjusted, and the error value shows a downward trend on the whole. After more than 200 training rounds, the error value is basically stable.

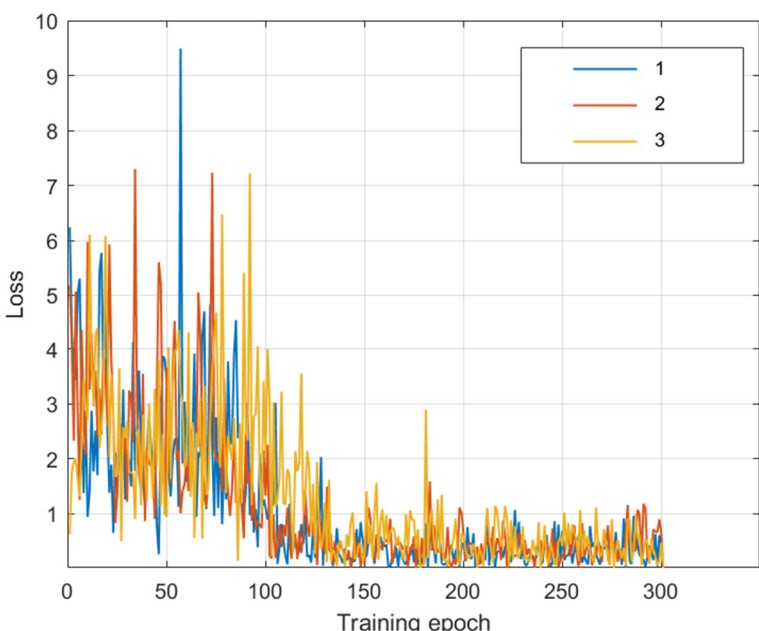

**Figure 10.** Learning curves in different scenarios.

### 4.2.2. Comparison of the Algorithm Performance

The performance of different algorithms is compared in this section. Considering that the input data are damaged by the target signal, the other parameter settings are the same as in scenario 1. Two hundred Monte Carlo experiments were used to calculate the average output SINR, where the optimal SINR is calculated on the premise that all information is fully known and the test data are not affected by the target. The performance of different algorithms versus SNR and snapshot is shown in Figures 11 and 12. It can be noticed that the relative ADMM algorithm has a performance second only to the C-ADMMN algorithm, which is the basis of the advantages of the algorithm. The utilization of the trained C-ADMMN yields better beamforming performance and consistently has the best performance throughout the process of SNR and fast beat number changes. In addition,

under different SINR and fast beat conditions in the experiment, the trained network in scene 1 was used. For different target and interference situations, the output SINR is the same as Figure 11 and is not drawn repeatedly here. This shows that the trained network has a good universality.

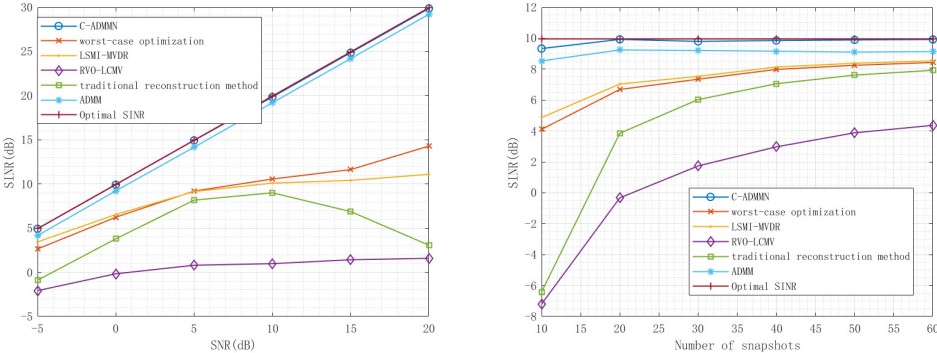

**Figure 11.** Outputs SINR versus SNR and snapshots in scenario 1.

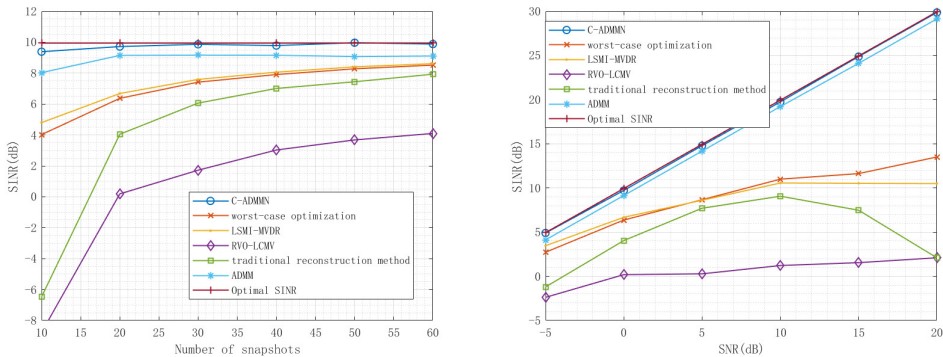

**Figure 12.** Outputs SINR versus SNR and snapshots in scenario 3.

For an intuitive presentation, the RMSE and operation times of different algorithms in different scenarios are given in Table 1. Table 1 provides a more intuitive perspective, indicating that the proposed C-ADMMN algorithm substantially reduces operation time and exhibits minimal deviation from the theoretical optimal value.

**Table 1.** Performance comparison of different algorithms in different scenarios.

|         | Algorithm              | RMSE    | Time        |
|---------|------------------------|---------|-------------|
| Scene 1 | worst-case optimization | 3.7511  | 48.845960   |
|         | LSMI-MVDR              | 3.4097  | 51.199925   |
|         | RVO-LCMV               | 10.1474 | 48.082712   |
|         | ADMM (max 300)         | 0.8192  | 35.036440   |
|         | C-ADMMN (30 layer)     | **0.4594** | **3.770667** |
| Scene 2 | worst-case optimization | 9.8828  | 49.784180   |
|         | LSMI-MVDR              | 10.3290 | 53.399999   |
|         | RVO-LCMV               | 18.7958 | 47.921498   |
|         | ADMM (max 300)         | 0.8969  | 33.197317   |
|         | C-ADMMN (30 layer)     | **0.4173** | **3.607241** |
| Scene 3 | worst-case optimization | 3.8176  | 49.784180   |
|         | LSMI-MVDR              | 3.6098  | 50. 870276  |
|         | RVO-LCMV               | 10.1997 | 48.201499   |
|         | ADMM (max 300)         | 0.9078  | 33.372376   |
|         | C-ADMMN (30 layer)     | **0.5211** | **3.496087** |

## 5. Conclusions

This paper proposes a robust adaptive beamforming method based on deep-unfolding networks, which adopts the form of deep-unfolding networks for the subspace recovery algorithm based on atomic norm optimization. The backpropagation process within the deep-unfolding networks is employed to optimize hyper-parameters during the traditional atomic norm optimization iteration. Moreover, the proposed method determines the optimal hyper-parameters under different interference noise matrix conditions, enhancing the performance of the traditional interference subspace recovery method based on the ADMM algorithm.

**Author Contributions:** Conceptualization, Z.G. and X.Z.; methodology, M.R.; writing—review and editing, X.S.; supervision, Z.L. All authors have read and agreed to the published version of the manuscript.

**Funding:** This research was supported in part by the National Natural Science Foundation of China (62022091, 61921001).

**Data Availability Statement:** Not applicable.

**Conflicts of Interest:** The authors declare there are no conflicts of interest.

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
