# Peer review of "ADMM-Net for Beamforming Based on Linear Rectification with the Atomic Norm Minimization"

_remotesensing, doi:10.3390/rs16010096_

Round 1

Reviewer 1 Report

Comments and Suggestions for Authors

The proposed algorithm presents an innovative approach to adaptive beamforming in radar signal processing, addressing challenges associated with interference sources and corrupted training data. By combining ADMM with a deep unfolding network, the algorithm achieves efficient parameter optimization, enhancing adaptability and performance. The simulations demonstrate that the algorithm outperforms traditional methods like RVO-LCMV, maintaining strong main and side lobe features while effectively inhibiting interference. The approach's ability to handle varying interference and noise conditions, along with its universal applicability across different scenarios, showcases its robustness. Overall, the report introduces a promising solution to improve beamforming accuracy and efficiency in complex radar environments.

Author Response

Dear Prof.,

We sincerely appreciate your careful reviewing our manuscript entitled “ADMM-net for beamforming based on linear rectification with the atomic norm minimization” [ID: remotesensing-2786261]. Those comments and suggestions are all valuable and quite helpful for improving our paper, as well as the important guiding significance to our researches. We have studied the comments carefully and have made some revisions which we hope meet with approval.

We hope that the revised manuscript can meet with approval and your further advice is also much appreciated.

Reviewer 2 Report

Comments and Suggestions for Authors

In this paper, the authors proposed an interesting method of adaptive beamforming which deep unfolded a jamming subspace recovery algorithm. The idea is novel and the simulation results seem convincing. A few matters need to be addressed.

1. The structure of the network and description of the method need to be explained in detail, especially how to train the network.

2. The authors need to emphasize and explain the importance of the hyper-parameters in the paper. After all, this is the reason of using deep unfolding technique.

3. Some minor typos need to corrected. For example, in page 5, “By directly minimizing the number of atoms is an NP problem,…”. The authors need to examine their manuscript more carefully.

Author Response

Dear Prof.,

We sincerely appreciate your careful reviewing our manuscript entitled “ADMM-net for beamforming based on linear rectification with the atomic norm minimization” [ID: remotesensing-2786261]. Those comments and suggestions are all valuable and quite helpful for improving our paper, as well as the important guiding significance to our researches. We have studied the comments carefully and have made some revisions which we hope meet with approval.

  1. We have further refined the description of the network structure, enriching it with relevant information for each section. The network training section was also supplemented accordingly to the suggestions, including the type of loss function, the selection of the optimizer, and the setting of the learning rate.
  2. We have already described the importance of hyperparameters at the end of Section 3.1, and accordingly lead to later questions about the network design.
  3. We have corrected this issue and reviewed the full manuscript carefully.

Reviewer 3 Report

Comments and Suggestions for Authors

In this paper, a novel adaptive beamforming named complex alternating direction method of multipliers network, has been proposed. This method isolates the interference subspace with deep unfolding an ADMM algorithm. The main conception is quite interesting and novel, however some issues still bother me.

1.       In equation (9), is the direction of target  assumed to be known? If so, the authors need to explain how to acquire such information in practical radar systems.

2.       Some of the references are too old. The authors should include more latest advances in adaptive beamforming.

3.       I wish to see more simulation results. How does the method perform in different SNR scenarios and different snapshots scenarios?

Comments on the Quality of English Language

No comments

Author Response

Dear Prof.,

We sincerely appreciate your careful reviewing our manuscript entitled “ADMM-net for beamforming based on linear rectification with the atomic norm minimization” [ID: remotesensing-2786261]. Those comments and suggestions are all valuable and quite helpful for improving our paper, as well as the important guiding significance to our researches. We have studied the comments carefully and have made some revisions which we hope meet with approval.

  1. The target position in space is typically estimated using the direction of arrival algorithm, which serves as the foundation for beamforming. Relevant instructions have been added to the manuscript.
  2. We added more up-to-date references as suggested.
  3. According to the suggestion we added the simulation results when the target angle is transformed (for case 3 condition) and further analyzed the simulation results.